# Identifying metabolic adaptations characteristic of cardiotoxicity using paired transcriptomics and metabolomics data integrated with a computational model of heart metabolism

**Bonnie V. Dougherty**[1], **Connor J. Moore**[1], **Kristopher D. Rawls**[1], **Matthew L. Jenior**[1], **Bryan Chun**[1], **Sarbajeet Nagdas**[2], **Jeffrey J. Saucerman**[1], **Glynis L. Kolling**[1,3], **Anders Wallqvist**[4], **Jason A. Papin**[1,3,5]*

**1** Department of Biomedical Engineering, University of Virginia, Charlottesville, Virginia, United States of America, **2** Department of Microbiology, Immunology, and Cancer Biology, University of Virginia Health System, Charlottesville, Virginia, United States of America, **3** Department of Medicine, Division of Infectious Diseases and International Health, University of Virginia, Charlottesville, Virginia, United States of America, **4** Department of Defense Biotechnology High Performance Computing Software Applications Institute, Telemedicine and Advanced Technology Research Center, U.S. Army Medical Research and Development Command, Fort Detrick, Maryland, United States of America, **5** Department of Biochemistry & Molecular Genetics, University of Virginia, Charlottesville, Virginia, United States of America

* papin@virginia.edu

**Data Availability Statement:** Code to reproduce this analysis is available at (https://github.com/

## Abstract

Improvements in the diagnosis and treatment of cancer have revealed long-term side effects of chemotherapeutics, particularly cardiotoxicity. Here, we present paired transcriptomics and metabolomics data characterizing *in vitro* cardiotoxicity to three compounds: 5-fluoro-uracil, acetaminophen, and doxorubicin. Standard gene enrichment and metabolomics approaches identify some commonly affected pathways and metabolites but are not able to readily identify metabolic adaptations in response to cardiotoxicity. The paired data was integrated with a genome-scale metabolic network reconstruction of the heart to identify shifted metabolic functions, unique metabolic reactions, and changes in flux in metabolic reactions in response to these compounds. Using this approach, we confirm previously seen changes in the p53 pathway by doxorubicin and RNA synthesis by 5-fluorouracil, we find evidence for an increase in phospholipid metabolism in response to acetaminophen, and we see a shift in central carbon metabolism suggesting an increase in metabolic demand after treatment with doxorubicin and 5-fluorouracil.

## Author summary

Improvements in the diagnosis and treatment of cancer have revealed long-term side effects of chemotherapeutics, particularly cardiotoxicity. Here, we compare the cardio-toxic effects of 3 drugs, 5-fluorouracil, acetaminophen, and doxorubicin, using

BonnieDougherty/Cardiotoxicity). A list of DEGs found in this analysis can be found at (https://github.com/BonnieDougherty/Cardiotoxicity/tree/main/data/RNA-seq/DEGs). FASTQ files are available under GEO166957 available at (https://www.ncbi.nlm.nih.gov/geo/query/acc.cgi?acc=GSE166957).

**Funding:** Support for this project was provided by the United States Department of Defense (W81XWH-14-C-0054 to JAP), the National Institutes of Health (NIH grant HL137755 to JJS, DK132369 to JAP), and the National Science Foundation Graduate Research Fellowship Program (awarded to BVD). The opinions and assertions contained herein are the private views of the authors and are not to be construed as the official or as reflecting the views of the U.S. Army or the U.S. Department of Defense. This manuscript has been approved for public release with unlimited distribution. The funders had no role in study design, data collection and analysis, decision to publish, or preparation of the manuscript.

**Competing interests:** The authors have declared that no competing interests exist.

transcriptomic and metabolomic data. By integrating these data into a genome-scale metabolic network reconstruction of the heart, we were able to identify metabolic adaptations in response to cardiotoxicity that avoided detection in traditional gene enrichment and metabolomic techniques. Using this approach, we confirm known mechanisms of doxorubicin-induced cardiotoxicity and provide hypotheses and potential mechanisms for metabolic adaptations in cardiotoxicity for 5-fluorouracil, doxorubicin, and acetaminophen. We hope future work using our novel paired transcriptomic and metabolic characterization of *in vitro* cardiotoxicity can help improve the current network heart model and further characterize the role of chemotherapeutics in cardiotoxicity.

## Introduction

Multiple chemotherapeutics have been identified as increasing the incidence of cardiovascular events, both in the short- and long-term following treatment, now termed cardiotoxicity [1]. It is now well-established that multiple chemotherapeutics are associated with adverse cardiovascular events, such as left ventricular dysfunction and chronic heart failure [1]. However, mechanisms of cardiotoxicity are not well understood. Recently, changes in glucose uptake have been noted to precede clinical measures of heart dysfunction in both spontaneously hypertensive rats [2] and in patients undergoing chemotherapy with known cardiotoxic drugs [3–4], suggesting that broad metabolic adaptations to drug treatment may yield insight into mechanisms of cardiotoxicity.

Genome-scale metabolic network reconstructions (GENREs) provide an opportunity to mechanistically connect changes in metabolites with changes in transcriptomics, identifying potential mechanisms for the production of metabolites. GENREs provide a mechanistic representation of cellular metabolism, including the stoichiometric coefficients for metabolic reactions and the connectivity between genes and the individual reactions they govern. Previous studies have used transcriptomics data with metabolic network reconstructions to study metabolic adaptations in hepatotoxicity [5–7] and nephrotoxicity [8–9], though the study of cardiotoxicity using metabolic network reconstructions is currently unexplored.

In the current study, we integrate paired transcriptomics and metabolomics data with a novel heart-specific genome-scale metabolic network (GENRE) [10] to predict metabolic adaptations in cardiotoxicity. Paired transcriptomics and metabolomics data were collected for primary rat neonatal cardiomyocytes exposed to three compounds: 5-fluorouracil (5FU), acetaminophen (Ace), and doxorubicin (Dox). Both Dox and 5FU were selected based on their established cardiotoxicity [11–12] while Ace was chosen based on previous studies exploring hepatotoxicity and nephrotoxicity [5,8]. Furthermore, Dox has multiple hypothesized mechanisms of toxicity [11] whereas 5FU does not have established hypotheses for mechanisms of cardiotoxicity [12]. Integrating multiple forms of omics data with functional models of metabolism yields unique insight into potential metabolic adaptations in cardiotoxicity. Through this integrated approach, we (a) recapitulate published metabolic adaptations in 5FU and Dox cardiotoxicity, (b) propose new metabolic adaptations in Ace cardiotoxicity, and (c) propose the role of shifts in key metabolic reactions as representative of increased metabolic demand in 5FU and Dox in response to the primary chemotherapeutic mechanisms of action.

## Results

### Optimizing concentrations of compounds to characterize in vitro cardiotoxicity

Given that most toxicity studies have limited rationale for their chosen drug concentrations, we aimed to deliberately choose cardiotoxic concentrations of 5-fluorouracil (5FU), acetaminophen (Ace), and doxorubicin (Dox) for our *in vitro* studies. For the purposes of this study, cardiotoxic concentrations were defined as the concentration that elicited a significant decrease in cell viability without a significant increase in cell death compared to the control at 24 hours (Fig 1). 10 mM 5FU and 2.5 mM Ace elicited a significant decrease in cell viability at 24 hours without a significant increase in cell death. 1.25 µM Dox induced a significant decrease in cell viability but also a significant increase in cell death at 24 hours (Fig 1).

As with previous studies [5,8], paired transcriptomics and metabolomics data were collected at both 6 (S1 Fig) and 24 hours (Fig 1) after drug exposure to capture early and late toxicity. There was no significant increase in cell death for any compounds for our chosen concentrations at 6 hours. There was a significant decrease in cell viability for the chosen concentration of Ace at 6 hours, indicating that Ace has an early but sustained effect on cell metabolism in cardiomyocytes (S1 Fig).

### Unsupervised machine learning of transcriptomics and metabolomics data highlights underlying drug-induced shifts in cellular activity

In order to identify the largest sources of variability between the conditions, individual samples for either transcriptomics (transcript counts) or metabolomics (metabolite abundances) data

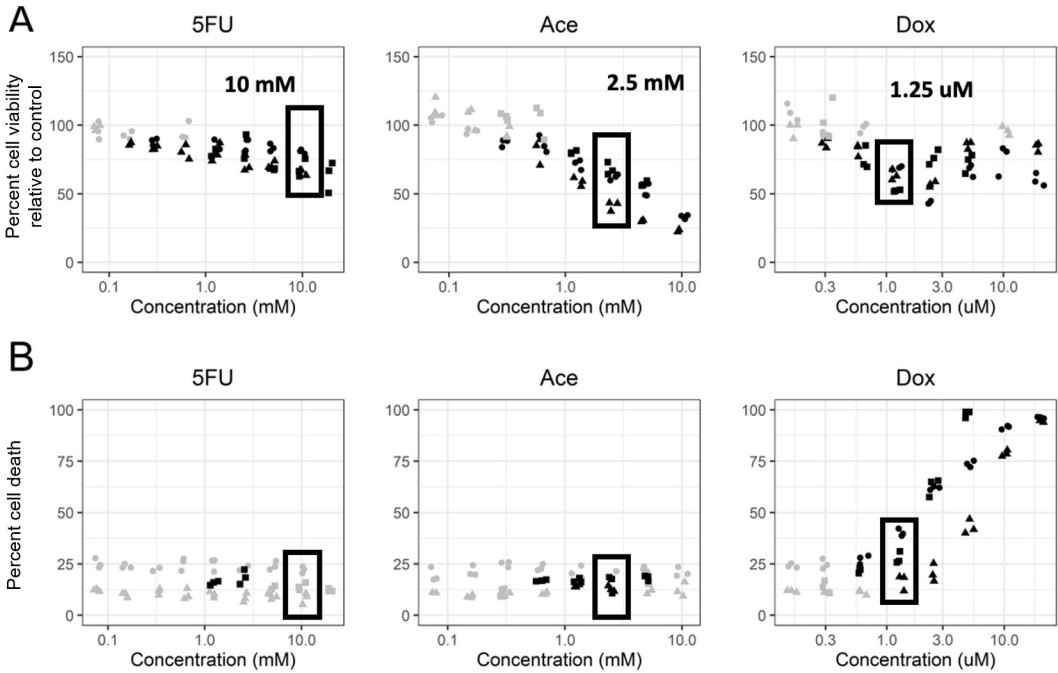

**Fig 1. Choosing cardiotoxic drug concentrations that elicited a change in metabolism without a significant increase in cell death.** (A) Percent cell viability with respect to controls was measured for a range of concentrations for the chosen compounds. The shapes represent different biological replicates. Black dots indicate a statistically significant change from the control condition, determined using Dunnett's test with a p-value < 0.05. Boxes indicate the concentrations chosen for the experimental studies. (B) Percent cell death calculated using a PI/Hoescht stain. The shapes represent different biological replicates. The black dots indicate a statistically significant change from the control condition, calculated using Dunnett's test with p-value < 0.05.

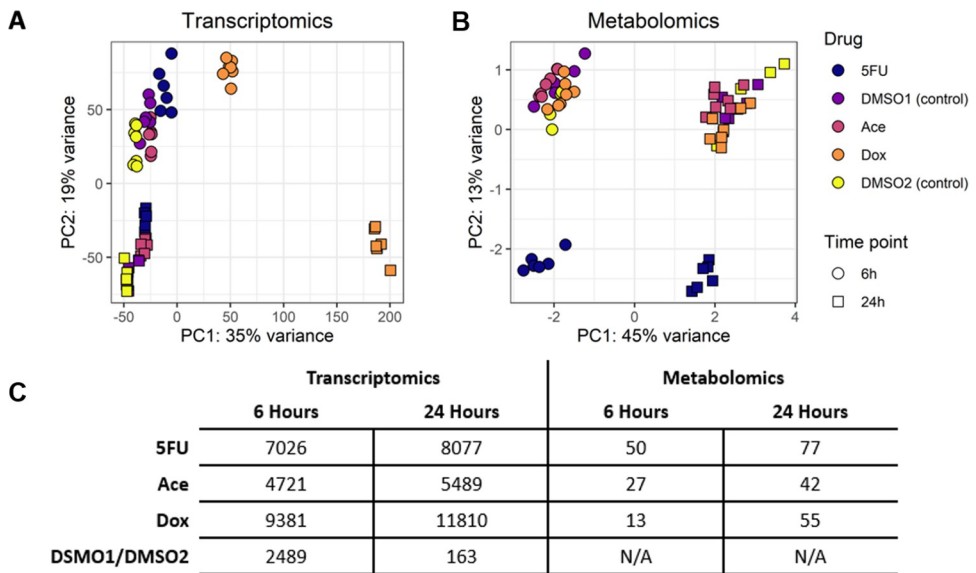

**Fig 2. PCA of transcript counts (A), PCA of scaled metabolite abundances (B), and table of protein-coding DEGs (C) for the three compounds and two DMSO concentrations at 6 and 24 hours.** (A) PCA of transcript counts separate by time and treatment condition, specifically the Dox treatment. (B) PCA of scaled metabolite abundances separate by time and treatment condition, specifically the 5FU treatment. (C) Quantification of the number of differentially expressed genes and differentially changed metabolites for each condition including between DMSO1 and DMSO2. DEGs were genes with an FDR < 0.01 and differential metabolites were metabolites with an FDR < 0.1.

were clustered in an unsupervised fashion using Principal Component Analysis (PCA) (Fig 2). For reference, the 5FU condition is paired with the DMSO1 control (1% DMSO) and the Ace and Dox conditions are paired with the DMSO2 control (0.25%) (Methods). For the transcriptomics data, within each time point, the control conditions (DMSO1 and DMSO2) separate from the treatment groups (5FU, Dox, and Ace) (PERMANOVA for treatment vs control, p-value < 0.001 for both 6 and 24 hour timepoints) (S2A Fig). The first principal component separates by the Dox treatment and the second principal component separates samples by time, 6 versus 24 hours. A gene enrichment analysis of the top 100 genes in the first principal component using the Hallmark pathways from the Molecular Signatures Database (MsigDb) [13–14] identified the p53 pathway as the only significantly enriched pathway, suggesting that Dox induces a unique response through the p53 pathway compared to the other compounds. The main function of the p53 pathway is to monitor cell division, including DNA replication [15]. Two of the proposed chemotherapeutic mechanisms of action of Dox are intercalation with DNA and the inhibition of topoisomerase II [16]; both mechanisms would therefore elicit a response from the p53 pathway. The second principal component separates between the 6-hour and 24-hour samples, including the control samples, suggesting a potential adaptation to culture conditions. A gene enrichment analysis for the top 100 genes in the second principal component identifies Myc targets as uniquely enriched, where consistent decreased expression of the Myc pathways is consistent with a transition from the fetal gene program [17] and increased expression consistent with a response to cardiomyocyte stress [18–19].

A PCA of log-scaled and median-centered metabolite abundances separates in the first principal component by time and in the second principal component by the 5FU treatment (Fig 2B). As with the transcriptomics data, the clear separation by time point suggests a potential adaptation of the primary cells to *in vitro* culture conditions. A bi-plot (S2B Fig) of the top 10 metabolites responsible for the PCA separation identifies erythritol, a derivative of glucose

metabolism [20], and ethylmalonate, a branched chain fatty acid, for primary separation in the first principal component, suggesting a change in glucose and fatty acid metabolism over time, though it is unclear in which direction this change occurs. The second principal component is separated by the 5FU treatment. Given that the chemotherapeutic mechanism of action of 5FU is acting as an analogue for uracil and interfering with RNA synthesis [21], it is not surprising to see clear separation in the metabolomics data for cells treated with 5FU. A bi-plot (S2B Fig) of the top 10 metabolites responsible for the separation identifies uracil, phosphate, and 2-deoxyuridine as primarily driving separation in the second principal component, indicating changes in uracil synthesis and general metabolism as separating 5FU from the other conditions. However, in contrast to the transcriptomics results, there is no clear separation among any conditions, except for 5FU treatment. This lack of separation could result from the number and type of metabolites that were profiled or could suggest a more nuanced change between the Dox and Ace conditions and their respective controls. Finally, the number of differentially expressed genes (DEGs) and differentially changed metabolites were quantified for each treatment and time point (Fig 2C, S1 Table). We see the least amount of DEGs in the DMSO concentration comparison, but it does appear to have a short-term effect on transcription at 6-hours. As was expected from the PCAs, the Dox condition has the largest number of DEGs at both 6 and 24-hours while the 5FU condition has the largest number of differentially changed metabolites at the 6 and 24-hour conditions.

## Gene enrichment and metabolomics data identify common signatures of toxicity but cannot readily identify mechanisms of cardiotoxicity

The large number of DEGs for each condition necessitates an enrichment approach to identify changed pathways that are shared across conditions and time points and could therefore be suggestive of a common mechanism of cardiotoxicity. Pathway enrichment analysis was performed using the 50 Hallmark gene sets defined in the MsigDB [13] (Fig 3A). Given the large number of DEGs, it was necessary to use both a lower FDR cutoff to define differential expression and a higher p-value cutoff for enriched gene sets; genes were considered differentially expressed with an FDR $< 0.01$ and gene sets were enriched with a BH-adjusted p-value $< 0.1$. Consistent with their known mechanisms of chemotherapeutic efficacy, the 5FU and Dox conditions are enriched for genes related to DNA repair at both the 6 and 24-hour timepoint. As with the PCA data, the p53 pathway is enriched for the Dox condition at both 6 and 24 hours as well as enrichment in at least one timepoint for the other two conditions. Finally, genes related to oxidative phosphorylation are enriched, particularly at 6 hours, for both 5FU and Dox, consistent with changes in cellular metabolism. Two pathways are enriched across all conditions at 6 hours, the E2F targets and G2M checkpoint, both of which are pathways involved in the cell cycle. Only one pathway is enriched across all three conditions at 24 hours, the epithelial to mesenchymal transition (EMT) pathway.

Next, metabolites were identified that were significantly changed across all conditions within a timepoint (Fig 3B). We see increased production of free phosphate for all three compounds, suggesting a significant change in metabolism for the chosen concentrations, consistent with the measured decrease in metabolic activity at the chosen concentrations (Fig 1). The 24-hour 5FU and Dox treatments show an increase in 2'-deoxyinosine production, a behavior previously seen in the presence of reactive oxygen species (ROS) stress [22]. In these groups, increased 2'-deoxyuridine is also observed, suggesting changes in DNA damage and uracil metabolism [23–24]. Metabolites were both (a) differentially consumed between conditions when compared to blank media controls (S3A Fig) and (b) produced or consumed between conditions when compared to blank media controls (S3B Fig). Of note is uracil; in the 5FU

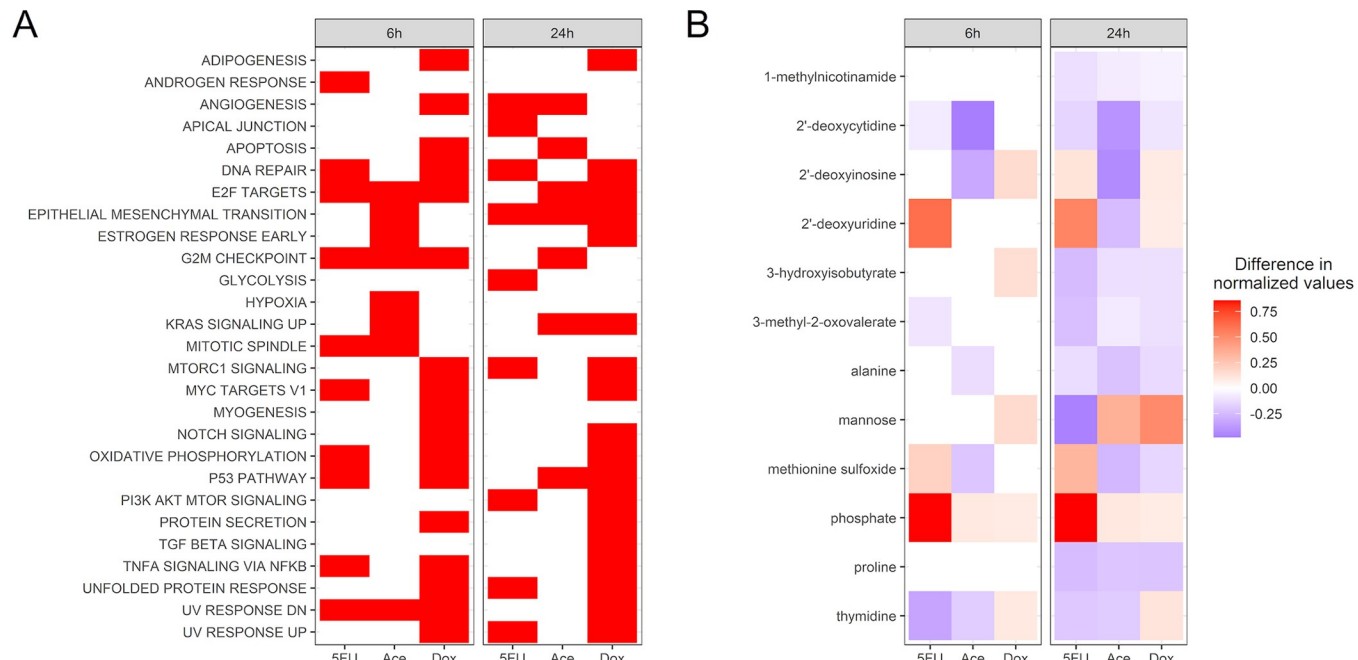

**Fig 3. Identifying biomarkers of toxicity from the transcriptomics and metabolomics data sets.** (A) Enrichment analysis for genes with an FDR < 0.01 using the Hallmark gene sets from the Molecular Signatures database. A red box indicates enrichment with a p-value < 0.1. Only gene sets that were enriched in at least condition are displayed here. (B) Metabolites that were identified to be significantly changed in production between the treatment and control (Mann-Whitney, FDR < 0.1). The color represents the mean difference in normalized metabolites between the treatment and control, where a negative value indicates decreased production and a positive value indicates increased production.

treatment condition at both timepoints, uracil was consumed more in the treatment compared to the control condition but in the Dox, Ace, and control conditions at 24 hours, uracil was produced. A metabolite that had a measured change in production between treatment and control (Fig 3B) can serve as a potential biomarker of *in vitro* cardiotoxicity; however, a specific mechanism for the change in production, and its relationship to changes in consumption, is difficult to assess. Further, there are no available methods for connecting the DEGs and differentially changed metabolites to identify metabolic adaptations during cardiotoxicity and potential genes driving those adaptations. Metabolic network reconstructions provide an opportunity to connect the measured changes in the transcriptome with the measured changes in the metabolome to identify these potential metabolic adaptations during cardiotoxicity.

## Reconstruction of a rat-specific heart GENRE from an existing human-specific heart GENRE

Paired GENREs of human (*iHsa*) and rat (*iRno*) metabolism have been published, which capture species-specific metabolic functions and species-specific gene-protein-reaction (GPR) rules [25]. However, integrating the data collected here requires a rat, heart-specific metabolic network reconstruction, which has not been published to date. A human-specific heart GENRE, *iCardio* [10], was recently published that was built from the paired human and rat GENREs, *iHsa* and *iRno*. *iCardio* was built by integrating tissue-specific protein expression data available in the Human Protein Atlas (HPA) (v18.proteinstlas.org; [26]) with the *iHsa* model followed by manual curation using pre-defined metabolic tasks. Given that *iHsa* and *iRno* were generated in parallel, the reactions included in the human-specific heart model, *iCardio*, directly map to a rat-specific heart model using the common reaction identifiers.

After including these reactions, 13 rat-specific reactions were added from the general rat model, *iRno*, that had literature evidence for expression in the heart or were necessary for metabolic functions (S1 File).

Next, the metabolomics data was used to identify metabolites that were measured in the collected dataset and included in the metabolic model. 121 metabolites mapped between the metabolomics dataset and metabolites in the model with many of these metabolites showing significantly differential abundance after drug treatment (S5A Fig). 75 of these metabolites had associated exchange reactions in the general rat model, indicating that the metabolite could be either consumed or produced in the model. From these exchange reactions, 37 were added to the rat-specific heart model from the general model to ensure that constraints could be placed for either production or consumption. Manual curation is often a time-intensive process and here we demonstrate the value of metabolomics data in identifying new potential reactions that should be added to tissue-specific models of metabolism.

Before integrating the transcriptomics with the model, we first identified the number of DEGs and differentially changed metabolites that the new rat-specific heart model captures (S4A Fig). The model captures ~10% of the DEGs across all treatments and time points and between 40–65% of the differentially changed metabolites such as pyruvate and glucose. Next, in order to confirm that the metabolic genes captured in the model are still capturing the underlying variability in the data (Fig 2A), PCA was used to identify the largest sources of variability in the metabolic genes represented in the model (S4B Fig). As with the previous PCA (Fig 2), there is a similar spread in the data. Using a similar enrichment approach as before, both the glycolysis and hypoxia Hallmark pathways, among others, were enriched in the top 100 genes separating the first principal component, suggesting that a change in glycolysis may have a unique role in doxorubicin toxicity. For the second principal component, the Hallmark pathways of cholesterol homeostasis, glycolysis, oxidative phosphorylation, and fatty acid metabolism, among others, were enriched. Again, this result suggests a difference in how these cells use glycolysis and fatty acid metabolism to produce ATP, supporting our previous data as well (S2B Fig). However, as with the previous results, it is unclear from the enrichment analysis alone what is the direction of change, i.e., from the fetal gene program, as is seen in maturing cardiomyocytes, or towards the fetal gene program, as has been noted in heart failure [27]. Nonetheless, the model captures metabolic adaptations in response to drug treatment. However, as with the previous analysis, general pathway-level changes do not yield insight into potential mechanisms of cardiotoxicity.

## Integrating transcriptomics data with a rat-specific, heart GENRE predicts novel metabolic functions altered in in vitro cardiotoxicity

Gene enrichment analyses can be helpful in identifying broad changes in a data set. However, as noted above, it can be difficult to understand the relationship between DEGs and metabolic shifts, even with measured metabolomics data. Here, the Tasks Inferred by Differential Expression (TIDEs) approach [10] was used to identify metabolic functions that are associated with a significant change in gene expression using the rat-specific heart model (Fig 4). The TIDEs approach uses complex GPR rules which describe the relationship between proteins in a reaction, such as isozymes (OR relationship) and protein complex subunits (AND relationship). We use these rules with a metabolic model to assign weights based on the log fold change of DEGs to individual reactions necessary to complete a metabolic task, such as production of ATP from glucose. Metabolic tasks were identified that are significantly associated with either increased or decreased gene expression in each drug condition. Statistical significance is calculated by randomizing log fold change values to create a distribution of metabolic task scores to

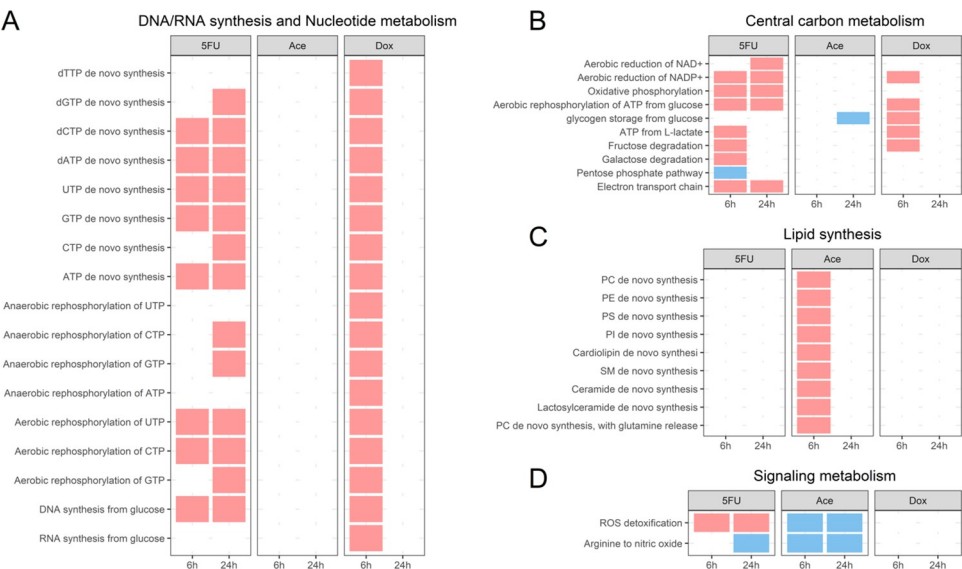

**Fig 4. TIDEs analysis reveals distinct changes in metabolic function in response to compounds.** A red box indicates significantly higher associated gene expression and a blue box indicates significantly lower associated gene expression. (A) Metabolic tasks related to DNA and RNA synthesis or nucleotide metabolism. (B) Metabolic tasks related to central carbon metabolism. (C) Metabolic tasks related to lipid membrane synthesis. (D) Metabolic tasks related to signaling metabolism.

assign significance to the original score. Using this approach, metabolic functions that are significantly associated with changes in gene expression are used to identify metabolic adaptations that are a potential cause or result of cardiotoxicity.

The log fold change of metabolic genes with an FDR < 0.01 were overlayed onto the rat-specific heart GENRE (Methods). Metabolic tasks with a p-value < 0.1 were identified as differentially changed. All metabolic tasks related to DNA/RNA synthesis and nucleotide metabolism are increased in the Dox condition at 6 hours (Fig 4A). In contrast, 5FU has metabolic tasks related to production of nucleotides and nucleoside triphosphates (NTPs) increased at both 6 and 24 hours, except for anaerobic phosphorylation of NTPs at 6 hours. Together, changes in both NTPs and dNTPs metabolism suggest that both 5FU and Dox have an effect on both RNA and DNA synthesis, which were expected in Dox due its enrichment of p53-related genes. These effects on nucleotide metabolism could serve as a potential metabolic stress on the cell. Additionally, we see several uracil-related tasks to be increased as well including UTP de novo synthesis and aerobic rephosphorylation of UTP, supporting our finding that uracil is differentially utilized in response to 5FU and Dox treatment (Figs 3B, S3 and S4).

Next, given that the Hallmark pathway of oxidative phosphorylation was enriched from the Hallmark gene set (Fig 3A), we examined changes in metabolic tasks related to central carbon metabolism (Fig 4B). A significant number of tasks were changed for the 5FU condition at both timepoints, indicating an overall strong metabolic shift in response to 5FU. The decrease of the metabolic task for the pentose phosphate pathway suggests that metabolic intermediates may be diverted to serve the increased metabolic demands induced by the 5FU treatment. As with the previous set of metabolic tasks, Dox is associated with significant changes in central carbon metabolism at 6 hours but no significant changes at 24 hours.

Given that Ace is not a known cardiotoxic compound, we next identified that metabolic tasks related to lipid membrane synthesis were uniquely increased in the Ace condition at 6 hours (Fig 4C). Particularly, metabolic tasks were upregulated for a variety of phospholipids,

suggesting a unique metabolic adaptation in response to Ace treatment. Of particular interest, the metabolic task for cardiolipin synthesis, a key component of the mitochondrial metabolism, is significantly increased. One mechanism of Ace hepatotoxicity is associated with significant lipid peroxidation in response to increased oxidative stress [28], suggesting a potential shared mechanism between hepatotoxicity and cardiotoxicity.

Finally, metabolic tasks that were significantly changed in signaling metabolism were identified (Fig 4D). Here, we find the synthesis of nitric oxide from arginine was decreased for three of the six conditions, and we observe differences in ROS detoxification between 5FU and Ace. Given the known role of Ace ROS production in Ace hepatotoxicity [28], it is interesting to see decreased gene expression for the metabolic task of ROS detoxification and the pentose phosphate pathway. Decreased synthesis of NO has been noted in heart failure [10,29–30], suggesting a shared metabolic marker of heart dysfunction.

Across all conditions and time points, the fewest significantly changed metabolic tasks were seen in the Dox 24-hour condition, which had the largest number of DEGs. For the metabolic task of arginine to nitric oxide and ROS detoxification, the underlying distribution of randomized task scores used to determine statistical significance was examined (S5 Fig). From these data, the Dox condition has a higher overall task score, indicating a higher overall average gene expression across reactions, but also has a wider spread for the underlying distribution of randomized task scores as a result of the large number of DEGs.

## Combined omics datasets predict novel metabolic adaptations in toxicity through integrated network-based analysis

Pathway-level analysis provides one point of view for interpreting changes in gene expression. However, pathways do not work independently, but rather, act in a coordinated effort to maintain cell function. GENREs are able to capture this relationship by determining flux through reactions in a model while meeting an objective function that represents a hypothesis for cell function. Here, the transcriptomics data was integrated with the rat-specific heart model using the RIPTiDe algorithm [31] to determine the reactions and fluxes that met the constraints provided by our objective function. A single objective function for non-proliferative cells is hard to define; here, the objective function of ATP hydrolysis, representing the ATP generated for cardiomyocyte contraction, as well as requiring minimal synthesis of DNA and RNA, was used (Methods).

Using the condition-specific transcriptomics data, condition-specific models were generated for each condition, both treatment and control groups. Details for the condition-specific models are in S2 Table. Non-metric multidimensional scaling (NMDS) was used to visually display the 50 flux samples for each condition in an unsupervised fashion (S6 Fig). For this approach, fluxes were plotted for the 87 reactions that are shared amongst all conditions. The NMDS plots would suggest that, as was suggested in earlier PCA plots, the DMSO controls each elicit a different metabolic response.

Supervised machine learning with random forest feature selection was used to identify the reactions that most distinguish the active metabolism of treatment from the respective control conditions. This analysis highlighted that every treatment and time point contained at least one reaction involved in central metabolism as highly distinguishing, suggesting unique divergent pathways for ATP production between treatment and control conditions. Second, when aggregating treatment conditions versus the control conditions, reactions were identified that were uniquely associated with changes in the treatment conditions. Flux samples for one of the reactions identified as highly distinguishing (Fig 5) demonstrates a difference in pyrimidine metabolism across treatment groups. In the case of 5FU and Dox, decreased flux is predicted

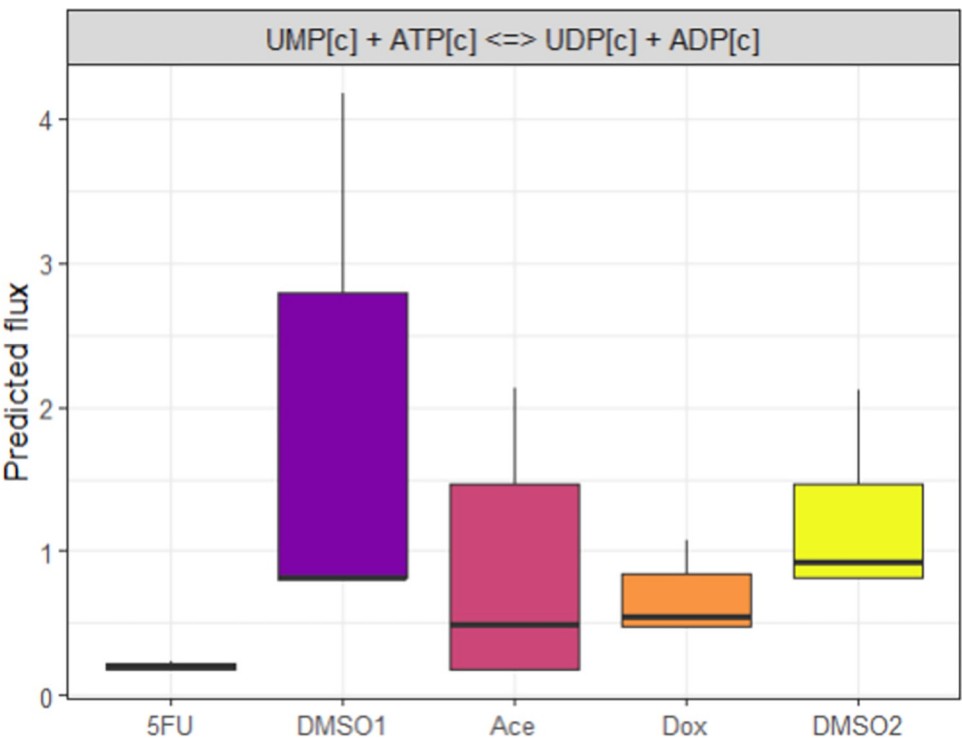

**Fig 5. Condition-specific models integrating the metabolomics and transcriptomics 24 hour data identify unique reaction common across treatment groups that are driven by changes in transcription.** Distribution of predicted fluxes through one reaction that were predicted to most strongly predict the difference between the treatment and control conditions.

in the production of UDP from UMP (Fig 5), where ATP is used to facilitate this reaction. The predicted flux for the change in RNA-related metabolism in the presence of these drugs again highlights their primary chemotherapeutic mechanism of DNA/RNA damage. It is important to note that the same constraints were placed on the overall network structure in this case (i.e., the same level of production of DNA and RNA). However, the transcript data imposed another constraint which, in this case, identified this reaction as different in all three treatment conditions.

## Discussion

Here, we present paired transcriptomics and metabolomics data characterizing *in vitro* cardio-toxicity for three compounds: 5-fluorouracil, acetaminophen, and doxorubicin. Here, cardio-toxic concentrations were defined as concentrations that induce a change in metabolism without a significant change in cell death for all three compounds. These concentrations were also found to be at or above previously measured serum concentrations for these drugs [32–34], indicating that they are physiologically relevant, but more work needs to be done to describe the role of pharmacokinetics and other factors to determine relevant concentrations to assess in cardiotoxicity studies. The data provides the first, to our knowledge, paired tran-scriptomic and metabolic characterization of *in vitro* cardiotoxicity. For the transcriptomic data, the compound Dox elicits the strongest transcriptomic response. Response was also strongly influenced by time. The time dependency could be time from primary isolation or time of drug exposure, an open question yet to be resolved. For the metabolomics data, time elicits the strongest metabolic response, followed by the 5FU condition. As with the

transcriptomics data, the influence of time could be time from primary isolation or time of drug exposure. In the metabolomics data, in contrast to the transcriptomic data, no difference in conditions outside the 5FU treatment was observed, indicating either a lack of sufficient coverage to identify metabolic differences or more nuanced changes in Dox and Ace and their respective controls. This result suggests that treatment may elicit significant changes in gene expression, while these transcriptomic effects may not result in direct changes to metabolism.

The treatment was confirmed using enrichment analysis, demonstrating significant enrichment for DNA repair in both 5FU and Dox, where both chemotherapeutic mechanisms of action target DNA or RNA synthesis. Additionally, metabolic changes were seen through enrichment in oxidative phosphorylation and increased release of phosphate measured in the metabolomics data across all conditions. The only Hallmark pathway that was enriched across all conditions at 24 hours was the EMT pathway. EMT is an important regulator in the development of the heart and, more recently, has been shown to have a role in the development of cardiac fibrosis [35]. Here, enrichment in genes in the EMT pathway could suggest a role for EMT in cardiotoxicity. While pathway level and metabolite changes are helpful, there are few methods currently available that use known mechanisms to model changes at the gene level to propose potential metabolic adaptations in response to treatment.

While standard differential gene expression analyses and differential metabolite analyses are useful, computational models of metabolism, such as those derived from GENREs, provide an opportunity to give mechanistic insight into the relationship between changes in expression and metabolism. These approaches provide a two-fold perspective by first (a) reducing the set of DEGs and reactions down to key metabolic reactions, and (b) providing a direct relationship between a prediction (e.g., a metabolic task or flux) and gene expression. Here, a key driver of the separation seen in the gene expression data (Fig 1) is metabolically related genes (S6B Fig), which was not identified in standard enrichment analyses. Second, we identify a metabolic reaction that serves as distinguishing between treatment and control conditions (Fig 5).

Consistent with the enrichment analysis, integration of DEGs with the TIDEs approach specifically identified changes in metabolic tasks related to DNA/RNA synthesis but also specifically identified the role of all nucleotides in both Dox and 5FU. In the case of Ace, the TIDEs approach identified increased synthesis of multiple phospholipids as metabolic adaptations during cardiotoxicity. Lipid peroxidation of membrane phospholipids is one proposed mechanism of acetaminophen-induced hepatoxicity [28] but has not been proposed in cardiomyocytes. Nitric oxide synthesis and ROS detoxification were identified as common metabolic functions altered across conditions. Nitric oxide synthesis is a proposed mechanism of doxorubicin-induced cardiotoxicity [36] but has not yet been identified for 5FU cardiotoxicity. Here, decreased NO synthesis and changes in ROS detoxification were identified as early and shared markers of *in vitro* cardiotoxicity. Finally, it is important to note the difference in number of TIDEs identified between conditions, where the Dox 24-hour condition has the lowest number of TIDEs. Given that there are the largest number of DEGs in the Dox condition, most genes that were included in the TIDEs analysis for the Dox 24-hour condition were differentially expressed, meaning that a metabolic function had to have a large gene expression signature to be identified as significant (S6 Fig). This is one weakness of the current approach. Efforts to develop future approaches should work towards being independent of the number of DEGs and could instead rely on transcript counts in the treatment and control conditions to determine shifts in metabolic functions.

Although enrichment analyses are helpful in determining broad metabolic changes, it is important to view these changes in the context of the entire metabolic network. The RIPTiDe analysis pipeline results in predictions of reaction flux distributions that satisfy the given objective function and transcriptomics constraints. In this way, RIPTiDe also provides additional

resolution at the reaction-level that may not be accessible in subsystem-level analyses. It is important to highlight that the chosen objective function, synthesis of ATP, DNA, and RNA, utilizes only central and key metabolic reactions. More complex objective functions provide the opportunity to explore additional, peripheral metabolic pathways. However, it is difficult to define appropriate objective functions for human metabolism. Therefore, for this study, we chose to focus on the core metabolic functions of ATP synthesis, representing energetic demands for cardiac contraction, and baseline DNA and RNA synthesis.

Random forest analysis identified reactions that were shared among individual conditions but whose flux separated between conditions. These reactions represent changes in flux that are necessary either for ATP production, metabolite production, or DNA/RNA synthesis that differ between conditions, indicating divergent pathways of flux. Across all three treatment groups, reactions for central metabolism were identified as significantly different between conditions, indicating a baseline divergent flux for ATP production. When comparing all treatment conditions to the control conditions, reactions related to the creation of UDP from UMP using ATP were significantly more predictive when using random forest analysis, particularly for the 5FU and Dox conditions, suggesting these chemotherapeutics may be altering baseline pyrimidine metabolism.

Together, the paired transcriptomics and metabolomics data taken with predictions made using the rat-specific model of metabolism provide mechanistic insight that was not clear from either set of data on its own. For Dox, we identified shifts in metabolic tasks related to nucleotide metabolism, ROS detoxification, and NO synthesis, consistent with previously published hypotheses for mechanisms of toxicity [37–38], as well as shifts in nucleotide metabolism as a potential additional metabolic stress contributing to cardiotoxicity. For 5FU, we identified shifts in metabolic tasks related to nucleotide metabolism, ROS detoxification, and NO synthesis, and, as with Dox, shifts in nucleotide metabolism suggesting increased metabolic stress from the chemotherapeutic mechanism of action of 5FU. Finally, for Ace, we identified shifts in metabolic tasks related to lipid synthesis.

There are several important limitations to these analyses. DMSO has been shown to decrease the ATP production of treated cells [39] which may have impacted which concentrations are chosen for our analysis. DMSO has also been shown to alter the methylome of cells [40], and we see transcriptomic differences between the two DMSO concentration, particularly at the 6-hour timepoint. These effects may interfere with the analyses performed here, and these conclusions could differ when studied *in vivo*. Additionally, by only including DEGs that are directly related to metabolism in our analysis, we ignore the contribution of a large proportion of the DEGs to the toxicological response. There are certainly additional important mechanisms that will need to be considered in future work. We also recognize that our methods for combining these data are not the only possible approaches, and additional analyses with tools like IMPaLA [41] could provide additional insights that we do not have here. For our metabolic modeling analysis, we use synthesis of ATP, DNA, and RNA as the objective function. Since this is not the sole biological objective of a cardiomyocyte, this analysis could miss mechanisms that impact other cellular tasks such as biomass growth.

Future work is necessary to trace pathway fluxes to determine how fluxes through these individual reactions influence other parts of metabolism. The collected metabolomics data also serve as an additional set of constraints to place when integrating the transcriptomics data with GENREs. However, more work needs to be done in developing methods to identify the relationship between a metabolic constraint and a change in predicted flux. In addition, the provided paired transcriptomics and metabolomics data provide a starting point for improvements to the present metabolic network reconstruction. A number of metabolites were measured as produced but could not be produced with the individual condition-specific models,

either because of missing internal reactions or missing constraints. Finally, future work can explore the use of additional objective functions that replicate proposed mechanisms of toxicity, such as increased ROS production or synthesis of key cellular proteins, which may provide further explanation for the measured changes in the metabolomics data. We hope that the novel data and results in this study provide the foundation for these additional investigations and can allow for the further research of mechanisms of cardiotoxicity.

## Methods

### In vitro culture conditions

Primary neonatal rat cardiomyocytes were isolated and cultured according to previously published protocols [42]. After the initial plating, cells were maintained for ~36 hours in plating media containing low glucose DMEM and M199 supplemented with L-glutamine, Penicillin-Streptomycin, 10% horse serum and 5% FBS. Cells were serum starved overnight (~12 hours) before running experiments in serum free, ITSS-supplemented plating media. Cells were observed to beat spontaneously within 24–48 hours after isolation, confirming metabolic activity and functionality.

For experiments to determine optimal drug concentrations, cells were seeded in 96-well plates at a density of 100k cell/well. The initial range of concentrations used for 5FU, Dox, and Ace (Tocris) were selected based on previous studies for Dox [43], 5-FU [43–44], and Ace [45]. Drug stocks were prepared according to manufacturer's instructions using sterile DMSO and were diluted in plating media before treatment. Concentrations that induced cardiotoxicity were determined using parallel measures of cell death and cell reducing potential, referred to as cell viability in this manuscript (10 mM 5FU, 2.5 mM Ace, 1.25 μM Dox). Cell death was determined as the number of propidium iodide (PI) positive cells divided by the total number of Hoescht positive cells. Fluorescence data from treated cells was background corrected using blank wells before using CellProfiler [46] to segment nuclei and measure fluorescence intensity for both PI and Hoescht. In the case of doxorubicin, which is fluorescent at overlapping wavelengths with PI, wells containing the respective concentrations of doxorubicin were used for background subtraction. Measures were aggregated from four fields of view for each drug concentration. Cell viability, which measures cell reducing potential and thus cell metabolism, was measured using the RealTime-Glo MT Cell Viability kit (Promega, Catalog #G9711) which has been done successfully in the past [47]. This kit uses a substrate that can permeate into the cell, and when reduced, it fluoresces, allowing us to quantitively measure the capacity of the cell to metabolically reduce metabolites. Both measures were repeated for three separate wells, representing three technical replicates for each condition, as well as on three separate days using different primary cell isolations, representing three biological replicates for each condition. Statistical significance was calculated using Dunnet's t-test [48] which accounts for the dose-dependent nature of the data. A p-value $< 0.05$ was considered statistically significant.

### RNA isolation, sequencing, and analysis

For the paired transcriptomics and metabolomics data, hearts were harvested in parallel from three litters of rats on the same day. After parallel digestion, cells were mixed from all isolations before plating in 12-well plates at a density of 1.2 million cells/well. For reference, two separate DMSO controls were run (DMSO1 at 1% DMSO for the 5FU condition; DMSO2 at 0.25% DMSO for the Ace and Dox conditions). Primary rat neonatal cardiomyocytes were exposed to the chemicals mentioned above at the chosen concentrations for either 6 or 24 hours. After exposure, as has been done in previous studies [5,8], the cells were lysed with Trizol to begin RNA extraction. Cell lysates were mixed with chloroform and spun in phase-lock

gel tubes inside a cold room and the upper phase was then decanted into new tubes. Isopropanol and glycogen were added to the mixture, incubated overnight at -20C and spun again resulting in an RNA pellet, which was washed with 75% ethanol twice. DNA was removed using the TURBO DNA-free kit (Invitrogen, #AM1907) and then RNA was quantified using the QuBit RNA Broad Range detection kit (Invitrogen, #Q10210). RNA was sent to GeneWiz (https://www.genewiz.com/en) for PolyA selection, library construction, and sequencing. RNA was sequenced using 2x150bp paired-end (PE) readings and fastq files were generated. Kallisto v 0.46.0 [49] was used to pseudo-align raw fastq files under default settings to the *Rattus norvegicus* Ensembl v96 transcriptome. Transcript abundances were then aggregated to the Entrez gene level in R v. 3.6.3 with the package tximport [50]. Genes with consistently low counts ($< 10$) across all samples were removed. Variance stabilization was performed with the vst() function from the limma package [51], and reads were normalized to library size using estimateSizeFactors from DESeq2 [52]. Differentially expressed genes (DEGs) were determined using DESeq2 [52] with a significance threshold of FDR $< 0.1$.

Principal component analysis (PCA) was performed using the variance stabilized gene counts [52] with the prcomp function in R. Statistical significance for the separation between treatment and control groups was calculated using adonis2 function in the Vegan package in R [53].

Following identification of DEGs, two approaches were used to identify pathways significantly changed in the data: enrichment using Hallmark gene sets from the Molecular Signatures Database [13] and Tasks Inferred from Differential Expression (TIDEs) for identifying differentially changed metabolic functions [10]. For the enrichment analysis, the enricher function from the clusterProfiler package was used with the top 100 genes in PC1 and PC2 as done previously [14]. Due to the large number of DEGs, enrichment was determined using genes with an FDR $< 0.01$ and pathways were defined as statistically significant with a p-value $< 0.1$ following Benjamini-Hochberg (BH) correction. For the TIDEs analysis, the subset of genes that mapped to the rat model and that had an FDR $< 0.01$ were used; pathways with a p-value $< 0.1$ when compared to randomly shuffled DEGs were defined as statistically significant.

## Collecting and analyzing metabolomics data

As described above, primary rat neonatal cardiomyocytes were exposed to the compounds at the chosen concentrations. Before lysing the cells with Trizol, the cell supernatant was removed and sent to Metabolon for analysis (https://www.metabolon.com/). Raw area counts were obtained for 181 named metabolites. Metabolites that had greater than 60% missing values (13 named metabolites) were removed from the data set. Next, missing values were imputed as half of the minimum raw area count within a metabolite. Values were then log-scaled and mean-centered within a metabolite. The Mann-Whitney U-test was used to determine if a metabolite was produced or consumed relative to the blank media samples (n = 3) as well as differences between treatment and control conditions. Metabolites were considered to be significantly changed if the p-value $< 0.1$ following Benjamini-Hochberg correction.

## Building a rat cardiomyocyte-specific metabolic model from the human cardiomyocyte-specific metabolic model, iCardio

The previously published human heart metabolic model, *iCardio* [10], was used to build a rat-specific heart metabolic model to contextualize changes in metabolites and DEGs. All reactions that were included in the human model were included in the rat model. The 6 updates that were made in the human model [10] were also made to the general rat model. Since each

model contains species-specific reactions, each of the 23 rat-specific reactions were manually curated to determine if they should be included in the heart; 13 were included based on literature evidence (S1 File).

Further curation was necessary based on metabolites that were measured to be either produced or consumed in the metabolomics data. Metabolites were mapped between the metabolomics data and the metabolic model using compound identifiers from the KEGG database. Exchange reactions, which are reactions in the model that transport a metabolite into or from the extracellular compartment, were added from the general rat model to the heart model if a metabolite was measured to be either consumed or produced relative to blank media in the metabolomics data. These reactions were necessary in order for a metabolite to be modeled as either produced or consumed. Reactions added back to the heart model from the general rat metabolic network are summarized in S1 File.

To identify shifted metabolic functions with the TIDEs approach, the developed rat-specific heart model was used with the previously published list of metabolic tasks [10]. Only genes that mapped to the metabolic model were included in the analysis. For each metabolic gene, a weight was assigned based on the FDR where a gene with an FDR $< 0.01$ was assigned its log fold change as a weight and 0 otherwise. As with the previous publication, reaction weights were determined based on the weights for individual genes in the complex gene-protein-reaction (GPR) rules. Task scores for individual tasks were calculated as the average weight across reactions in that metabolic task. To establish statistical significance, the weights for each metabolic gene were randomly shuffled 1000 times and significance was determined by comparing the original metabolic task score to random data.

## Predicting reaction metabolic flux using the RIPTiDe algorithm

The RIPTiDe algorithm [31] was used to integrate the transcriptomics data to identify the most likely flux distributions in the rat-specific heart model network for each of the 10 conditions, including both treatment and control groups at both time points. RIPTiDe identifies possible optimal flux distributions in a metabolic network given the cellular investments indicated by the transcriptomic abundance data. The media composition and metabolomics data were used to place constraints on metabolite consumption for each condition-specific model. Here, the cellular objective function was defined to be ATP hydrolysis, with an upper bound of 100 units of flux, and production of 1 unit of RNA and DNA, representing general cell maintenance. The exchange reactions for consumed metabolites were given a lower bound of -10, representing a theoretical overabundance of each metabolite for the given objective. These constraints allow for metabolites to be consumed at a relatively high flux. Finally, the upper bound of internal reactions was set as $10^6$ to ensure that internal fluxes were not constraining the solution space.

After applying these condition-specific constraints, condition-specific models were generated by integrating the median transcripts per million (TPM) for each gene within a condition using the RIPTiDe algorithm [31]. The RIPTiDe algorithm was run with a minimum fraction of 90% of the objective to ensure that differences in ATP flux were not the main determinants of differences between the condition-specific models. Each condition-specific model was flux-sampled 50 times to obtain a range of possible flux distributions that satisfied the pFBA assumption.

Following RIPTiDe analysis, flux samples for each condition were analyzed to identify both unique reactions for each condition and reactions whose flux separated between conditions [31]. Non-metric multidimensional scaling (NMDS) ordination of Bray-Curtis distances between flux samples, calculated using the Vegan package in R [53], was used to visualize

differences for reactions that were shared between all conditions. Finally, random forest feature selection [54] was used to determine the reactions whose fluxes most separated between each treatment and control group.

## Supporting information

**S1 Fig. Cell viability and cell death measures following 6 hours of exposure to compounds.** Shapes indicate different cell isolations. Black dots indicate a statistically significant change from the control condition (p-value < 0.05) calculated using Dunntt's test. Black boxes indicate the chosen concentrations for cardiotoxicity characterization. (A) Percent cell viability for a range of concentrations of treatment following 6 hours of exposure. (B) Percent cell death measured using a Hoescht/PI stain.
(TIF)

**S2 Fig. Additional PCA plots demonstrate (A) separation of treatment vs control groups and (B) the top 10 metabolites separating the PCA of the scaled metabolite abundances.** (A) PCA with the Dox samples removed demonstrates separation between treated and control samples at both 6 and 24 hours. (B) The top 10 metabolites show separation in both the first and second principal component. For the first principal component, ethylmalonate and erythritol have a strong influence, suggesting a phenotypic switch between fatty acid and glucose utilization over time, although the direction is unclear. For the second principal component, phosphate and uracil separate the 5FU condition.
(TIF)

**S3 Fig. Metabolomics data shows differential consumption and production of metabolites between blank media and control/treated samples.** (A) Metabolites that were measured to only be consumed across both drug treated and control conditions when compared to blank media. Here, red indicates a metabolite that was consumed more in the treatment condition than the control condition, blue indicates a metabolite that was consumed more in the control condition than the treatment condition, white indicates a metabolite that was consumed in either/both the treatment and control conditions but there was no difference in consumption and grey indicates a metabolite that was not consumed in either the treatment or control condition when compared to blank media. An example is shown for a metabolite that was more consumed between treated and control conditions (adenine) and a metabolite (hepatonate) that was consumed in a treated/control condition but there was no difference between the treated/control conditions. Black dots indicate a condition where there was a significant change in the metabolite when compared with blank media (Mann Whitney U-test, FDR < 0.1). P-values are shown for comparisons that were significant between a treatment and its respective control. (B) Metabolites that were measured to be both produced and consumed across conditions. Here, the red line indicates the mean value for the blank media samples. Black dots indicate conditions where there was a significant change when compared to blank media (Mann Whitney U-test, FDR < 0.1). A black dot above the red line indicates that a metabolite was produced whereas a black dot below the black line indicates a metabolite that was consumed. (C) Metabolites that were not significantly changed from blank media samples but were measured to change between treatment and control samples. Here, red indicates that a metabolite was present at a higher level in the treated group compared to the control group whereas blue indicates a metabolite that present at a lower level compared to the treated group. Given that there is no significant change with respect to the blank media, directionality of change (i.e. consumption or production) cannot be determined.
(TIF)

**S4 Fig. The rat-specific heart model captures changes in DEGs and metabolomics.** (A) The number of DEGs (FDR < 0.1) and differentially changed metabolites (FDR < 0.1) that map to the rat-specific metabolic mode. (B) A PCA of the normalized gene counts that map back to the rat-specific heart model demonstrate clear separation, similar to the PCA of all gene counts (Fig 2), confirming that metabolism has a large determinant in separating conditions.
(TIF)

**S5 Fig. Distribution of random task scores for the metabolic tasks for arginine to nitric oxide and ROS detoxification demonstrate the underlying distribution of DEGs.** A red background indicates a metabolic task associated with a significant increase in gene expression and a blue background indicates a metabolic task associated with a significant decrease in gene expression (p-value < 0.1). The red dashed line indicates the task score for the actual gene expression data whereas the black bars indicate the calculated task scores when the gene expression data is randomized. In this case, the distribution for the Dox data is significantly wider (i.e. larger range on the x-axis), indicating a larger overall absolute change in gene expression requiring a higher overall average gene expression for ROS production to be deemed significant and a lower overall average gene expression for arginine to nitric oxide to be deemed significant.
(TIF)

**S6 Fig. RIPTiDe models after integration with metabolomics and transcriptomics data.** NMDS of the 50 flux samples for each condition at (a) 6 hours and (b) 24 hours. For each flux sample, fluxes were only taken for the 112 reactions that are shared between all conditions.
(TIF)

**S1 Table. Differential expressed genes separated by treatment, time point, and direction of change.** This table clarifies which DEGs in Fig 2C are up-regulated and down-regulated.
(XLSX)

**S2 Table. RIPTiDe models after integration with metabolomics and transcriptomics data.** The number of reactions included in each model and the p-value for the Spearman correlation between reaction flux and transcript abundance.
(XLSX)

**S1 File. Rat-specific metabolic reactions mapped from the human *iCardio* model or manually added from *iRno* to ensure metabolic function.**
(XLSX)

**S1 Text. All supplementary information included in one document including figures, tables, and descriptions.**
(DOCX)

## Acknowledgments

We would like to thank Bethany Wissman for assisting with the isolation of the primary rat neonatal cardiomyocytes and Laura Dunphy for feedback on the manuscript.

Non-standard Abbreviations and Acronyms

(GENRE); Genome-scale metabolic Network Reconstruction

(GPR); gene-protein-reaction

(Dox); doxorubicin

(Ace); acetaminophen

(5FU); 5-fluorouracil

(OCR); oxygen consumption rate

## Author Contributions

**Conceptualization:** Bonnie V. Dougherty, Glynis L. Kolling, Anders Wallqvist, Jason A. Papin.

**Data curation:** Bonnie V. Dougherty, Bryan Chun, Sarbajeet Nagdas, Glynis L. Kolling.

**Formal analysis:** Bonnie V. Dougherty, Connor J. Moore.

**Funding acquisition:** Bonnie V. Dougherty, Jeffrey J. Saucerman, Jason A. Papin.

**Methodology:** Bonnie V. Dougherty.

**Visualization:** Bonnie V. Dougherty.

**Writing – original draft:** Bonnie V. Dougherty.

**Writing – review & editing:** Bonnie V. Dougherty, Connor J. Moore, Kristopher D. Rawls, Matthew L. Jenior, Bryan Chun, Sarbajeet Nagdas, Jeffrey J. Saucerman, Glynis L. Kolling, Anders Wallqvist, Jason A. Papin.

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
