## [Decision Letter · Decision Letter 0]

29 Apr 2023

Dear Jason,

Thank you very much for submitting your manuscript "Identifying metabolic adaptations of cardiotoxicity using paired transcriptomics and metabolomics data integrated with a computational model of heart metabolism" for consideration at PLOS Computational Biology.

As with all papers reviewed by the journal, your manuscript was reviewed by members of the editorial board and by several independent reviewers. In light of the reviews (below this email), we would like to invite the resubmission of a significantly-revised version that takes into account the reviewers' comments.

We cannot make any decision about publication until we have seen the revised manuscript and your response to the reviewers' comments. Your revised manuscript is also likely to be sent to reviewers for further evaluation.

Sincerely,

Pedro Mendes, PhD

Academic Editor

PLOS Computational Biology

Douglas Lauffenburger

Section Editor

PLOS Computational Biology

Reviewer's Responses to Questions

**Comments to the Authors:**

Reviewer #1: This manuscript deals with the survey of the transcriptomics and metabolomics changes occurring in primary cultures of neonatal rat cardiomyocytes exposed to the action of common oncologic drugs, namely 5-fluorouracil, acetaminophen, and doxorubicin. A network metabolic model was reconstructed from the gene expression data based on the human cardiomyocyte model and combined with the multi-omics data to gain mechanistic insights into the effects of and adaptations to these chemotherapeutic agents that lead to cardiotoxicity.

I have a series of comments and criticism that the authors should address. The present version of the manuscript is not satisfactory or appropriate for the broad readership of PLoS Computational Biology.

Lines 99-100. The authors should discuss how these concentrations compare to those achieved in serum during chemotherapy of cancer patients to evaluate whether the toxicity levels achieved in the cultures may occur in vivo during treatment.

Line 102. Does the denomination “cell reducing potential” refers to a property related to the redox potential? Because of the importance of the results the authors should specify how is “cell reduction potential” defined and how it is experimentally determined. A reference to a product kit is not enough.

Lines 178-185. The list of DEG that you are highlighting should be included at least as a supplementary table.

Line 187. Phosphate. Do you mean free phosphate or phosphorylated metabolites? Please clarify.

Lines 186-207 Regarding the bioinformatic analysis, the authors do not take full advantage of the “multi-omics” approach. An integrated multi-omics analysis of metabolomics-transcriptomics can be performed correlating observed changes with the purpose of determining which pathways share both significant transcripts and metabolites.

Lines 223-224. Do the 121 metabolites include those with changed abundance (as in table 2B) in response to the treatments with the oncologic drugs?

Lines 233-234. This is a real limitation since the GEMRE model ignores 90% of the DEGs that may be the ones providing important mechanistic information about the drug action and cellular adaptive responses that could deliver hints to avoid the cytotoxicity. This should be discussed in a limitations section.

Lines 257-262. With so many acronyms those sentences are extremely difficult to understand except for experts in GENRE and TIDE approaches. The authors should include a table with acronyms which are overabundant in the manuscript. When approaches are combined, the authors should refer to the biochemical or conceptual basis thereof instead of assuming the readers’ familiarity with specific jargon.

Lines 293-294. Unclear sentence please rephrase.

Line 239. Despite the name of the enzyme catalyzing the reaction, PEP under physiological conditions is never directly obtained from Pyruvate. Pyruvate kinase (PK, all isoforms) behaves as irreversible and in the boundary conditions of the model it should be stated as such. In those organs performing gluconeogenesis there is a carboxylation of pyruvate to oxalacetate followed by the PEPCK catalyzed reaction to circumvent the lack of reversibility of PK. Although adult hearts are devoid of gluconeogenic enzymes, their presence may be possible in neonatal cardiomyocytes. The authors should check this possibility and correct the model and Figure 5A.

Lines 315-316. The choice of ATP hydrolysis as objective function appears somewhat arbitrary. I believe this issue deserves a more elaborated discussion, including the dismissal of other possible functions in a limitations section.

Figures in the supplement are very confusing:

Supplemental Figure 1 shows that the cells do not produce ATP unless stressed by the oncologic drugs, since DMSO shows negligible ATP production flux as measured by the Seahorse stress test. This result is rather counterintuitive which means that the cardiomyocytes under normal culture conditions display the same basal and oligomycin sensitive respiratory flux. And only the intoxication of the cultured cells with the drugs stimulates the cells to synthesize ATP under culture conditions. This should be explained and interpreted. Perhaps the OCR tests can be repeated in the presence of a beta-adrenergic challenge.

Supplemental Figure 4 I believe the meaning of consumption of metabolites should be clarified. What does it mean to have a negative consumption? It would be much easier to interpret if the authors present the relative abundance and when describing the results mention that a negative change would correspond to consumption and positive to accumulation or production.

Reviewer #2: In their manuscript Dougherty et al. combine transcriptomics and metabolomics data on the same samples in order to study drug-induced in vitro cardiotoxicity with three commonly prescribed cancer therapies. Additionally, they use the data along with a genome-scale metabolic model in order to reveal mechanistic processes underlying drug-induced cardiotoxicity. They identified DEGs and differentially changed metabolites at two time points by comparing treated vs. DMSO control rat cardiomyocyte samples. They apply gene enrichment analysis and discuss recovered pathways in the context of toxicity. Next they build a rat heart GENRE by combining published models including their previously published iCardio model and identified known metabolic tasks as well as potential novel ones.

The manuscript describes an interesting combination of several omics data and in particular how transcriptomics data can be combined with a metabolic model in order to reveal changes in the metabolism. It describes a data analysis work flow, no novel algorithms are developed.

My comments are:

1. Novelty: The difference to their previous paper seems to be mainly in the application of their iCardio model (adapted to the rat) with novel drug treatment data. This could be interesting from a toxicogenomics aspect but I do not see much algorithmic or methods development. Please clarify.

2. Significance: Authors should better highlight what the novel insights of the approach and the potential for users in the field are. There are multiple transcriptomics studies on Dox that also report metabolic changes and identify – amongst others – altered metabolic processes in the context of oxphos or others. What are the novel findings? Please compare to some of these and show that the used approach is more powerful.

3. DMSO controls: DMSO has been shown to induce broad effects in the methylome of cells (e.g. PMID: 30874586) which in fact can exceed the effects of the treatments under study. How is this reflected by the different dosages for DMSO1 and DMSO2? I would suggest computing DEGs from these two controls and test what TIDEs and pathways are found by these DEGs. This would help in evaluating the results.

4. Data normalization: How was transcriptome and metabolome data normalized? This is a crucial point and it should be described in detail in the Methods section.

5. Differential expression (Methods, Figure 2C, page, 20ff): It would be good to have numbers for up- and down-regulated genes. Additionally, are all DEGs protein coding? The RNA protocol is based on poly-A, so I assume read outs also containing long non-coding RNAs and pseudogenes for example. Since the main interest here is on enzymes and metabolically relevant protein coding genes, I would suggest giving the respective numbers for this category.

6. Pathway enrichment analysis (page 7 and 21): Please define pathway enrichment in the Methods section. Since only DEGs are used for pathway enrichment, I assume authors use exact Fisher test? In the light of the comment above, please reduce DEGs to protein coding genes otherwise the statistics might be misleading.

7. Sarcomere effects: I wonder a bit that pathway enrichment gives so little information on sarcomere effects. For example, there are strong influences of Dox for muscle contraction and in fact clinical markers of cardiomyopathy, such as TNNT2, NEBL etc., are related to Dox toxic effects. It is a bit puzzling that no cardiomypathy disease gene set is fished by the enrichment. Or may be these are not included in the "hallmarks signatures". Please explain?

8. There are several tools that provide common enrichment analysis for genes and metabolites, for example the Impala web server (PMID: 21893519). It would be interesting to compare the analysis to the output of such an analysis.

9. Page 6, line 131: enrichment was computed for the top 100 genes; this is quite an arbitrary threshold. Please motivate.

10. Page 7, line 155: top 10 metabolites; same comment.

11. Page 6, line154: „]“ is missing.

Reviewer #3: The authors present paired transcriptomics and metabolomics data characterizing in vitro cardiotoxicity to three compounds: 5-fluorouracil, acetaminophen, and doxorubicin. The authors methodology identified metabolic adaptations in response to cardiotoxicity vis a vis standard gene enrichment and metabolomics approaches identify some commonly affected pathways and metabolites. The authors claim that shifted metabolic functions, unique metabolic reactions, and changes in flux in metabolic reactions and hence potential mechanisms for metabolic adaptation in response to drugs were identified by integrating the paired data with a genome-scale metabolic network reconstruction of the heart.

In general, the manuscript is poorly written with respect to the results not being very clearly and sufficiently presented. The discussion section is too verbose. All in all, the methods and results are novel and exciting enough to spark excitement, but require more than superficial mention.

At this point a recommend a major revision.

I have a few suggestions:

1) It would be nice if in the abstract the authors include some key results from the used integrated approach with respect to both the validation of legacy metabolic adaptations in 5FU and Dox cardiotoxicity, the main new metabolic adaptations in Ace cardiotoxicity and the proposed shifts in key metabolic pathways that show increased metabolic demand in 5FU and Dox elucidating primary chemotherapeutic mechanisms of action.

2) Although the authors mention a significant increase in OCR for ATP production for 10 mM 5FU and 1.25 μM Dox and refer the reader to the supplemental data figure, it would be beneficial to mention a fold change right here.

3) The authors claim that the only A gene enrichment analysis of the top 100 genes identified p53 pathway as the only significantly enriched pathway in DOX treated cells. Also, they identify ethylmalonate, a branched chain fatty acid, for primary separation in the first principal component, suggesting a change in glucose and fatty when cells are treated with 5FU. However none of these observations are clearly tied up through gene expression/metabolomic data analysis.

4) Although the Dox condition has the largest number of DEGs at both 6 and 24-hours while the 5FU condition has the largest number of differentially changed metabolites at the 6 and 24-hour conditions, there absolutely no discussion of these results in the text.

5) It is alsocunclear, how do the authors deduce the consistent increase in production of 2’-deoxyinosine 2’-deoxyuridine in the both the 5FU and Dox treated cells as a marker for ROS.

6) Although uracil was identified as differentially utilized or produced under the varying

treatment conditions at 24 hours, again there is no correlation made to DEGs.

7) The model captures ~10% of the DEGs across all treatments and time points and between 40-65% of the differentially changed metabolites. It would be useful to mention some key metabolites. The a shift from glycolytic metabolism to fatty acid metabolism as a source of ATP is not supported by data that was collected from the experiments.

8) There are many typographical and grammatical errors and words in between sentences suddenly start with a capital letter.

**Have the authors made all data and (if applicable) computational code underlying the findings in their manuscript fully available?**

Reviewer #1: **No: **The datasets used for the analysis are not accessible at all, neither the model built based on these datasets is available.

Reviewer #2: Yes

Reviewer #3: Yes

PLOS authors have the option to publish the peer review history of their article (what does this mean?). If published, this will include your full peer review and any attached files.

Reviewer #1: No

Reviewer #2: No

Reviewer #3: No
---

## [Decision Letter · Decision Letter 1]

30 Oct 2023

Dear Professor Papin,

Thank you very much for submitting your manuscript "Identifying metabolic adaptations characteristic of cardiotoxicity using paired transcriptomics and metabolomics data integrated with a computational model of heart metabolism" for consideration at PLOS Computational Biology.

As with all papers reviewed by the journal, your manuscript was reviewed by members of the editorial board and by several independent reviewers. In light of the reviews (below this email), we would like to invite the resubmission of a significantly-revised version that takes into account the reviewers' comments.

While there has been progress in the manuscript that satisfied reviewers #2 and #3, please address the remaining issues from reviewer #1.

We cannot make any decision about publication until we have seen the revised manuscript and your response to the reviewers' comments. Your revised manuscript is also likely to be sent to reviewers for further evaluation.

Sincerely,

Pedro Mendes, PhD

Section Editor

PLOS Computational Biology

Douglas Lauffenburger

%CORR_ED_EDITOR_ROLE%

PLOS Computational Biology

Reviewer's Responses to Questions

**Comments to the Authors:**

Reviewer #1: The new version of the manuscript by Dougherty and coworkers does not show a significant improvement over the original one. I would have expected that after six months the authors would have incorporated a more elaborate research work; instead, I found in their revised manuscript just some edits, rewritten text, and clarifications, at least concerning the points I originally raised.

Regarding the issue of the reversibility of the pyruvate kinase (PK), the authors have not addressed my concerns in the revised manuscript. The PK may under controlled circumstances (in vitro) operate in the reverse direction. However, as I said in my original report, the chances of this happening under physiological conditions are slim. The PK affinity constant for Pyruvate is in the order of 7 mM when actual levels of the metabolite in vivo in most mammalian cells rarely reaches 100 microM, making the reverse reaction rather unlikely or extremely slow. The evolutionary selection of the activities of pyruvate carboxylase and PEPCK highly expressed in gluconeogenic organs such as liver and kidney, confirm the need of an alternative path to overcome the slow or negligible reverse flow through PK. I was expecting to find the results with a model incorporating these activities in their formulation, but the authors did not even attempt to perform such a small but significant change in their modeling.

Regarding the very low rates of ATP production in the case of the control due to larger number of viable cells in the respiration assay is an indication of the lack of linearity of the assay with respect to the number of cells in the well. This is an essential experimental condition that should be optimized before any assay in this system can be performed, Otherwise, the assay is saturated in the wells with many cells (e.g., those with the control cells in the absence of cardiotoxic drugs) and is likely still in the linear range for the small number of cells in the “treated” wells. Certainly, the comparisons should be made between wells with similar numbers of functional cells, to avoid issues of saturation of the signal. Given the lack of linearity in the assay, the comparison between control and treated cells is meaningless, and those results should be discarded.

The answers to my original comments lack details e.g., about where to find the modified parts in the revised manuscript. In one case, the authors mentioned the list of Differentially expressed genes (DEG) in a website in GitHub in a directory where there are a large number of files, without even mentioning the file name of the document containing the requested information or including such information in the revised manuscript.

Concerning the comment with respect to the “cell reducing potential” I insist that the denomination is confusing, and, in that context, should be replaced by “cell viability”. However, the reagent used has been in the market for relatively few years and there are no reports about how senescent cells (that have been demonstrated to be generated by oncologic drugs, e.g. Prasanna et al, 2021 PMID: 33792717) will behave in terms of reducing the commercial reagent used to assess “viability”. Thus, I have strong reservations regarding the interpretation of those results.

The limitations section in the discussion is written in a rather defensive way than in acknowledgement of the real limitations of the conclusions reached. For example, when the authors state: “by removing 90% of DEGs from our analysis, we focus on only metabolically relevant genes” without any functional demonstration about the relevance of the included (~10%) or excluded (~90%) genes.

Reviewer #2: The authors have addressed my points sufficiently. The improved ms is suited for publication.

Reviewer #3: The authors have addressed all queries posed in the first review. They have included required sections in the text.

The article is now more cogent than the first version.

I thus recommend this article for publication.

**Have the authors made all data and (if applicable) computational code underlying the findings in their manuscript fully available?**

Reviewer #1: **No: **

Reviewer #2: Yes

Reviewer #3: Yes

PLOS authors have the option to publish the peer review history of their article (what does this mean?). If published, this will include your full peer review and any attached files.

Reviewer #1: No

Reviewer #2: No

Reviewer #3: No
---

## [Decision Letter · Decision Letter 2]

15 Feb 2024

Dear Professor Papin,

We are pleased to inform you that your manuscript 'Identifying metabolic adaptations characteristic of cardiotoxicity using paired transcriptomics and metabolomics data integrated with a computational model of heart metabolism' has been provisionally accepted for publication in PLOS Computational Biology.

Best regards,

Pedro Mendes, PhD

Section Editor

PLOS Computational Biology

Douglas Lauffenburger

%CORR_ED_EDITOR_ROLE%

PLOS Computational Biology

Reviewer's Responses to Questions

**Comments to the Authors:**

Reviewer #1: The authors have addressed my questions satisfactorily.

**Have the authors made all data and (if applicable) computational code underlying the findings in their manuscript fully available?**

Reviewer #1: None

PLOS authors have the option to publish the peer review history of their article (what does this mean?). If published, this will include your full peer review and any attached files.

Reviewer #1: No

---

## [Editor Report · Acceptance letter]

23 Feb 2024

PCOMPBIOL-D-22-01830R2 

Identifying metabolic adaptations characteristic of cardiotoxicity using paired transcriptomics and metabolomics data integrated with a computational model of heart metabolism

Dear Dr Papin,

I am pleased to inform you that your manuscript has been formally accepted for publication in PLOS Computational Biology. Your manuscript is now with our production department and you will be notified of the publication date in due course.

With kind regards,

Lilla Horvath
